# Estimating person-specific neural correlates of mental rotation: A machine learning approach

**Sinan Uslu**[1]*, **Michael Tangermann**[2], **Claus Vögele**[1]

**1** Department of Behavioural and Cognitive Sciences, University of Luxembourg, Esch-sur-Alzette, Luxembourg, **2** Donders Institute for Brain, Cognition and Behaviour, Radboud University, Nijmegen, The Netherlands

* sinan.uslu@uni.lu

## Abstract

Using neurophysiological measures to model how the brain performs complex cognitive tasks such as mental rotation is a promising way towards precise predictions of behavioural responses. The mental rotation task requires objects to be mentally rotated in space. It has been used to monitor progressive neurological disorders. Up until now, research on neural correlates of mental rotation have largely focused on group analyses yielding models with features common across individuals. Here, we propose an individually tailored machine learning approach to identify person-specific patterns of neural activity during mental rotation. We trained ridge regressions to predict the reaction time of correct responses in a mental rotation task using task-related, electroencephalographic (EEG) activity of the same person. When tested on independent data of the same person, the regression model predicted the reaction times significantly more accurately than when only the average reaction time was used for prediction (bootstrap mean difference of 0.02, 95% CI: 0.01–0.03, p < .001). When tested on another person's data, the predictions were significantly less accurate compared to within-person predictions. Further analyses revealed that considering person-specific reaction times and topographical activity patterns substantially improved a model's generalizability. Our results indicate that a more individualized approach towards neural correlates can improve their predictive performance of behavioural responses, particularly when combined with machine learning.

## Introduction

Neural correlates quantify the relationship between neurophysiological properties and behavioural variables. Many studies have investigated the neural underpinnings of mental rotation. The mental rotation task has frequently been used to invoke complex cognitive processes including visuospatial representations and visual working memory. It involves the judgement of rotational invariance based on objects rotated in space. Neuroscience techniques such as positron emission tomography (PET scan), functional magnetic resonance imaging (fMRI),

**Data Availability Statement:** The source code is available at https://github.com/UsluSinan/EEG-correlates-of-mental-rotation under the MIT license. It contains scripts to simulate random EEG data in the required format to run the analyses. The

collected data for this study contain sensitive information and cannot be shared publicly due to GDPR. Access will require approval from the Ethics Review Panel (erp-submissions@uni.lu) and the Data Protection Office (dpo@uni.lu) of the University of Luxembourg.

**Funding:** This work is part of the Doctoral Training Unit Data-driven computational modelling and applications (DRIVEN) and is funded by the Luxembourg National Research Fund under the PRIDE programme (PRIDE17/12252781). The funders provided support in the form of salary for author SU. The funders had no role in study design, data collection and analysis, decision to publish, or preparation of the manuscript.

**Competing interests:** The authors have declared that no competing interests exist.

and electroencephalography (EEG) have provided insights into task-ongoing brain activity [1]. There is evidence of increased bilateral activity in several cortical areas including posterior parietal and frontal regions when performing mental rotations [2]. More recent findings suggest that further differentiation of activation patterns based on stimulus characteristics such as angular disparity [3] and whether or not body parts were represented [4] may benefit the identification of neural activity important for spatial manipulation. Additionally, the dorsal fronto-parietal network has been proposed as a neural substrate connecting motor and cognitive processes [5]. These investigations based on PET scans and fMRI provide insights regarding the localization of neural processes, however to a lesser degree about when the processing occurred. To extend these findings, studies recording task-ongoing EEG activity to capture temporal dynamics have shown oscillatory activities to be involved in mental rotation and have notably demonstrated the suppression of alpha (8–13 Hz) and beta (13–22 Hz) oscillatory activity with increasing demands for cognitive processing (i.e., event-related desynchronization–ERD [6, 7]). This body of findings indicates that mental rotation invokes consistent changes in neurophysiological activity which are stable across participants.

In patients showing progressive behavioural and neurological decline, as for example in those with Huntington's disease, subtle impairments in visuospatial abilities found in early stages transition to significant differences in symptomatic stages when compared to healthy controls [8]. Furthermore, there is evidence that mental rotation performance and corresponding activity of its neural correlates change in the presence of clinical deficits [1]. Compared to healthy participants, patients diagnosed with major depressive disorder showed reduced mental rotation performance manifesting in both higher reaction times and error rates [9]. Additionally, the performance decreased proportionally to an increase in depressive symptom severity [10]. In schizophrenia, an increase in mental imagery accompanied enhanced visuospatial imagery but did not reliably translate to mental rotation task performance with some studies observing an increase and others a decrease in performance [11, 12]. After bilateral stroke, damaging the posterior parietal cortex, patients often show impairments in visuospatial attention and body awareness [13]. In attention-deficit hyperactivity disorder, transcranial direct current stimulation demonstrated that increasing cortical excitability in the right posterior parietal cortex improved the attentional orienting network compared to a sham control [14, 15]. In summary, mental rotation in individuals with clinical conditions has been found to be altered; the identification of neural substrates, therefore, may represent a promising approach for neurophysiological stimulation studies to finally restore impaired behavioural functionality.

Since early neurophysiological studies, which observed behavioural changes after brain injuries, researchers have applied various approaches to assess neural underpinnings of behaviour [16]. One approach to investigate the neurophysiological basis of cognitive performance involves the identification of differences in neurophysiological properties for individuals with superior and for those with inferior performance: the neural efficiency hypothesis emerged stating that for identical task demands individuals requiring less neural resources outperform those requiring more resources [17, 18]. To classify participants using a binary performance representation, methods applied include (a) the median split where the classification is based on the relative position of a participant's task performance to the median performance derived from all participants and (b) contrasting neurophysiological activity between the participants with the highest and lowest performances [17, 19, 20]. Another approach to extract neural correlates involves the investigation of the continuous association between behaviour and neurophysiological phenomena. Quantifying the link between neural activity and cognitive performance in a continuous manner revealed for instance that lower ERD in alpha oscillatory activity was related to faster mental rotation [21]. Furthermore, machine learning approaches

modelling behavioural responses as a function of the preceding EEG power in four to ten frequency sub-bands performed with a mean absolute error between 100 and 600 ms [22, 23]. The overwhelming majority of the studies investigating neural correlates commonly assumed generalizable associations between brain activity patterns and some task performance parameters, which are stable across participants [24].

In a recent application of advanced algorithms for extracting neural activity patterns of mental states, researchers have extended previous approaches by making generalizable predictions at an individual level [24]. When training individual models to classify mental states based on fMRI recordings, they found person-specific features accurately identifying brain states. More specifically, they demonstrated that accuracy decreased when classifying brain states using a model trained on the fMRI data from another person. For EEG studies, algorithms to identify associations between electrocortical activity and behavioural measures included the supervised spatial filtering methods Common Spatial Patterns (CSP) and Source POwer Comodulation (SPoC) [25]. These methods allow to obtain individual linear combinations of multi-channel EEG and magnetoencephalography recordings that have an increased signal to noise ratio compared to the original channels. As a common pre-processing step, frequency filter banks decompose the EEG signals into defined frequency bands [26, 27]. While CSP addresses classification problems, SPoC extends this approach to continuous labels. For example, individual spatial filters resulting from SPoC related oscillatory rhythms to motor task performance in a continuous manner [25, 28]. These approaches take person-specific characteristics into account and notably increase the generalizability of predictions by carefully avoiding overfitting.

To assess the commonly assumed generalizability of neural correlates across participants [2–5, 11] and to improve the predictive accuracy of reaction times using EEG signals, we present a machine learning approach to extract person-specific neural correlates of mental rotation. We recorded EEG activity in eyes-open resting state and when performing a mental rotation task with varying levels of difficulty (i.e., angular disparity). We then removed artifacts and extracted EEG features preceding correct responses in the mental rotation task. To quantify the relationship between EEG features and reaction time (of correct responses), we trained a ridge regression model and finally evaluated its performance based on data not used for training (i.e., the hold-out set). For feature importance, we estimated Shapley Additive exPlanations (SHAP) values [29] in the hold-out set and thereby measured the contribution of each feature to the final prediction. Finally, we discuss the relevance and the limitations of our findings for application in personalized neurophysiological interventions.

## Materials and methods

### Participants

We collected the data in the context of a larger study for which we invited 40 participants (25 female, mean age: 24,97 years, age range: 19–35 years) to single two-hour laboratory assessments at the University of Luxembourg between September and November 2022. For data collection, we pseudonymized every participant's data, the key to which only the research staff had access to. To approximate a homogenous sample regarding visuo-spatial working memory performance, only participants between 18 and 35 years of age were recruited. In addition, participants were required to have normal or corrected-to-normal vision, and no history of mental disorders or neurological conditions [30]. The Ethics Review Panel of the University of Luxembourg approved the study, and all participants gave their written informed consent prior to participation (ERP 20–068). For reimbursement, participants received gift vouchers worth 10€ per hour.

## Mental rotation task

As part of the study, we assessed the performance of participants in a computerised mental rotation task with a total of 192 trials [31]. We will make the implementation of the mental rotation task available upon reasonable request. Each trial consisted of (1) a fixation cross which appeared in the centre of the screen for a random duration between 1000 ms and 3000 ms, and (2) the two three-dimensional (3D) figures, which appeared either until a response was given or for a maximum duration of 7500 ms. We used the Lab-Streaming Layer (LSL) [32] to synchronize markers defining the onset of a new phase or event (i.e., fixation cross, 3D figures, response) with the EEG stream. The two figures depicted either mirrored or unmirrored 3D objects with varying degrees of angular disparity. We instructed participants to press the 'Y' key on a QWERTZ keyboard whenever the presented 3D figures were unmirrored (i.e., rotationally invariant) and to press the 'N' key otherwise (i.e., when the 3D figures were mirrored). For each participant we randomly sampled without replacement 192 object pairs out of a pool with a total of 384 object pairs and stratified the sampling by angular disparity (i.e., 0˚, 50˚, 100˚, and 150˚) and by rotational invariance (i.e., mirrored, unmirrored). We instructed participants to choose a strategy for responding (i.e., either slower and more accurate or faster and less accurate) and to stick to it throughout the task.

## EEG recording

To capture electrocortical activity we mounted 32 Ag/ACl electrodes according to the 10/20 system on the participants' head and referenced them to FCz. A BrainAmp system then amplified and digitized the signals with a resolution of 16 bit and a sampling rate of 1 kHz (Brain Products, Gilching, Germany). To stream the data and to synchronize input timestamps from other sources (e.g., keyboard, stimuli) we used the LSL protocol [32]. We then accessed the LSL from within Python 3.7.3 [33] using the PyLSL library version 1.14.0 [32]. We implemented custom scripts based on the MNE library version 0.23.0 [34] for offline EEG processing and used SciPy library version 1.7.3 [35] for machine learning. Prior to the EEG recording in the mental rotation task, we recorded a one-minute, eyes-open, resting-state segment prior to which we instructed participants to horizontally move their eyes during the first five seconds.

## Data analysis

The associated source code to reproduce the analyses is available at https://github.com/UsluSinan/EEG-correlates-of-mental-rotation. The chronological order of processing steps is summarised below (Table 1). We trained a ridge regression model to predict the reaction time of correct responses in the mental rotation task based on features extracted from the EEG data prior to the response (Fig 1). First, we pre-processed the EEG data which involved removal of artifacts, bandpass filtering, the epoching of the continuous EEG signal and the removal of epochs with a duration of less than 700 ms (for more technical details, see the subsection on pre-processing). We then split the remaining data chronologically into a training set (consisting of the first 75% of epochs) and a hold-out set (with the remaining 25% of epochs). For details on the EEG features used to train our models, see the subsection on feature extraction. Based on the training data we performed a three-fold sliding window cross-validation to optimize the regularization intensity lambda. Finally, we trained our model with the specified lambda using all training data and evaluated its generalizability on the hold-out set. For model interpretation we relied on SHAP values.

**Pre-processing.**   Based on the one-minute resting-state recording we identified independent components, which contain artifacts (e.g., muscular activity, eye movements). Prior to

**Table 1. Pseudo code of the data analysis performed.**

| |
| --- |
| FOR each participant in participants: |
| read mental rotation task-related EEG from disk |
| apply ICA solution from resting-state EEG |
| apply bandpass filters |
| FOR each signal in bandpass filtered signals: |
| epoch the signal |
| remove fixation-cross epochs |
| remove epochs with incorrect responses |
| remove epochs shorter than 700 ms |
| cut epochs into 0 to 500 ms |
| split epochs at 75% into training and hold-out set |
| FOR lambda in lambdas: |
| cross-validate EEG model given lambda and training set |
| select lambda with minimum mean absolute error |
| train EEG model with selected lambda |
| define RT model |
| write EEG and RT model to disk |
| FOR each participant in participants: |
| evaluate intra-individual performance |
| evaluate inter-individual performance |

This table summarises the processing steps in chronological order. Please see here for more details: https://github.com/UsluSinan/EEG-correlates-of-mental-rotation.

training the fastICA implementation of the MNE toolbox, we applied a high-pass filter at 1 Hz [36, 37]. After visually inspecting the components' topographical activities, power spectral densities and time courses, we removed components representing artifacts. Using the spatial filters obtained based on the resting-state recording, this recording but also the EEG signals obtained during the mental rotation task were cleaned by first zeroing out the corresponding columns in the unmixing matrix before applying the final unmixing matrix back to the raw, unfiltered EEG signals.

To prepare the extraction of band power features from the cleaned EEG signals, we applied ten one-pass, non-causal, zero-phase, Hamming-windowed finite impulse filters that extracted the central frequencies at 2, 6, 10, 14, 18, 22, 26, 30, and 34 Hz with a width of 4 Hz [26, 38]. Markers defining the start and end of each presentation phase in the mental rotation task yielded a total of 384 epochs (i.e., 192 epochs with a fixation cross presentation and 192 epochs with 3D figures presented). From these epochs we removed those during which (a) the fixation cross appeared, (b) the given response was incorrect, and finally (c) an insufficient amount of data was recorded (i.e., when participants responded after less than 700 ms). After (a) and (b) we removed the final 200 ms from each of the remaining epochs to reduce the impact of motor preparation. As the application of SPoC to extract spatial filters required epochs of equal size, we exclusively kept the initial 500 ms from each epoch (i.e., from 0 to 500 ms) and removed epochs with less data available. We chose the initial 500 over the last 500 ms per epoch as EEG microstates indicate that the initial period contains processes related to the encoding of visual information and mentally rotating an object [39]. After removing on average 15.3% of epochs with 3D figures presented due to missing or too early responses, this procedure yielded on average across all participants 163 epochs (min = 105, max = 186 epochs) with 500 x 32 samples per epoch.

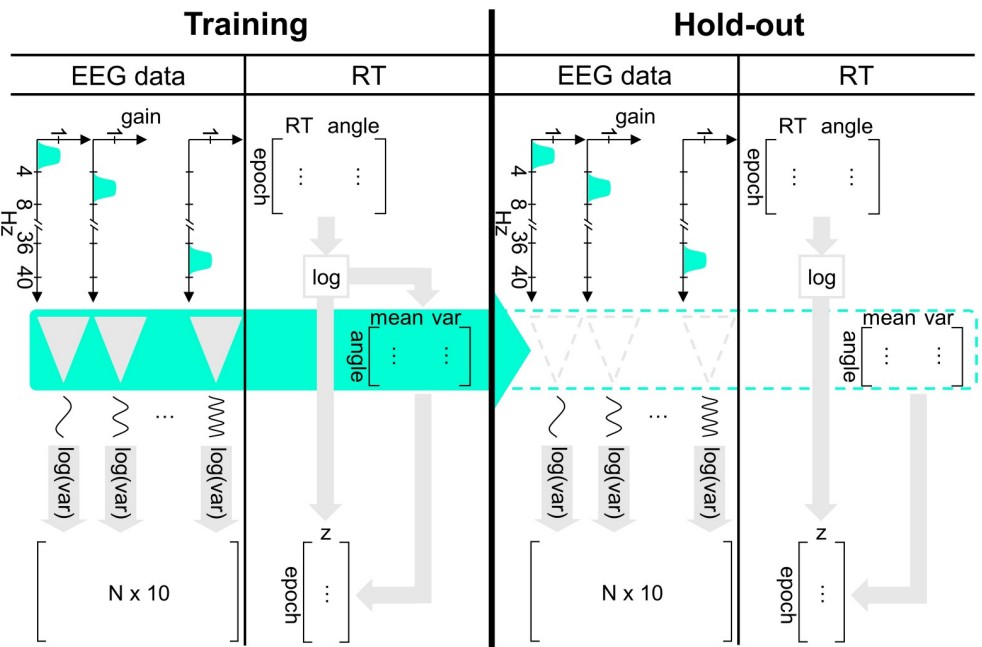

**Fig 1. Feature extraction and model evaluation.** For each participant, the first 75% of epochs were used as a training set and the remaining 25% as a hold-out set. In the training set, we first removed epochs which we classified as outliers (not visualized in the figure) and then extracted the EEG features by first bandpass and then spatially filtering the EEG signal and the labels by standardizing the log-transformed reaction times. We trained a ridge regression model to minimize for all epochs (N) the sum of squared errors between the reaction time labels and the model predictions with ten EEG features while penalizing the sum of squared coefficients with the optimized penalty term lambda. For the evaluation, we applied the same spatial and frequency filters and label standardization procedure to the hold-out set to extract new features and used the trained model to generate new predictions whose error we finally measured with the recorded reaction times in the hold-out set (see filled arrow in the figure).

**Feature extraction.**   To transform the recorded data into the same scale and to increase the signal to noise ratio, we applied a series of transformations to both the reaction time labels and the EEG signals to finally extract the EEG features based on bandpass filtered EEG signals (Fig 1). Due to their positively skewed distribution, we log-transformed the reaction times to approximate a Gaussian distribution. Exclusively in the training set we then calculated the median absolute deviation of log-transformed reaction time labels per angle (i.e., 0°, 50°, 100°, 150°) and removed on average 12 epochs which we considered as outliers (i.e., if the label was beyond a threshold of 2.5 x MAD [40]). This procedure yielded a final set of 110 epochs on average (min = 76, max = 138 epochs) with 500 x 32 samples each, which we used for training the regression models. Since the reaction time varied as a function of angular disparity and, hence, as a function of the degree to which mental rotation was involved to process the 3D figures, it was crucial for the interpretability of our model to take this relationship into account. For example, a reaction time of 800 ms may be considered as a) a relatively slow response if the 3D figures were rotated by 0 degrees, or b) a relatively fast response if the 3D figures were rotated by 150 degrees. To take the varying difficulty levels based on angular disparity into account, we standardized per participant the log-transformed reaction times for each angular rotation (S1 Fig). For hold-out and validation sets, we applied the means and standard deviations from the corresponding training set to standardize the reaction times, and to avoid data leakage.

For the EEG features, we started with ten bandpass filtered signals each of which consisted of 32 channels. For each frequency band (e.g., alpha), we then applied a spatial filter transforming the 32-channel signal into a univariate time series to increase the signal to noise ratio. Finally, we performed per frequency band and epoch (i.e., 500 EEG time points) a log-variance transformation of the univariate signal to approximate the band power, the final EEG feature, which yielded ten values per sample (Fig 1). By optimizing spatial filters via SPoC separately for each frequency band, the spatial filters adapted to the band-specific characteristics to maximize the comodulation between the spatially filtered EEG time-series and the reaction time labels. More specifically, we applied the $SPoC_\lambda$ algorithm [25] as implemented in the MNE framework (version 0.23.0), which maximizes the covariance between the two variables of interest (more details can be found in the documentation [41]). Given the relatively small number of epochs available for training, we only kept the component with the highest eigenvalue for each bandpass filtered multivariate EEG signal. Similar to the reaction times, we applied the same spatial filters, which we extracted from the training set to spatially filter the hold-out set in order to avoid data leakage.

**Hyperparameter tuning.** To calibrate the regularization parameter lambda towards minimization of the mean absolute error (MAE) for unseen data, we performed a chronological sliding window cross-validation procedure based on the first 75% of the available epochs (Fig 2). We applied a chronological cross-validation to take nonstationarities of the recorded EEG data into account [42] and defined overlapping windows to use most of the limited sample size for training. Importantly, while we re-used some of the preceding window's validation set to train the consecutive window's train-set, we did not re-use any sample for validation to reduce the risk of leakage (Fig 2). Candidate values for lambda were exponentially spaced between $10^{-1}$ and $10^4$ with each of which we trained and evaluated the model in all windows. In each window we trained the model with the initial 55% of epochs and evaluated its performance with the remaining 45% of epochs. For each lambda we stored the average of all three windows' MAE. Finally, the value for lambda with the lowest average MAE score, which was on median 241.13 (min = 3.53, max = 2476.37), was chosen for further processing.

**Final model evaluation and interpretation.** With the hyperparameter lambda minimizing the MAE in our hyperparameter tuning routine, we finally trained the model with the total training set (i.e., initial 75% of all epochs available) and evaluated its performance with the remaining hold-out epochs which had not been used yet (Fig 1). To estimate the ridge regression coefficients, we trained our models with the extracted features from the training set. Then, we predicted (log-transformed and standardized) reaction times using the trained model with the hold-out set's samples. To evaluate the model, we finally calculated the MAE between the predicted reaction times and the actual reaction times. For model interpretation we estimated SHAP values for each frequency band per epoch in the final hold-out set.

To evaluate the intra-individual prediction performance of the EEG model, we measured for each participant the MAE for (a) the "EEG model" which predicted reaction times based on the EEG features in the participant's hold-out set and (b) the "RT model" which estimated for the hold-out set the average, standardized reaction time per angular disparity (i.e., 0). The RT model did not take new data from the hold-out set into account and exclusively generated the new value based on the data from the training set. For the EEG model, we extracted the features from the last 25% of epochs using the same reaction time parameters (i.e., means and standard deviations per angular disparity) for standardization of reaction times and the same spatial filters for feature extraction from EEG data. Next, we estimated (a) reaction times with the trained model and the new data from the hold-out set (EEG model) and (b) average reaction times per angular disparity (RT model).

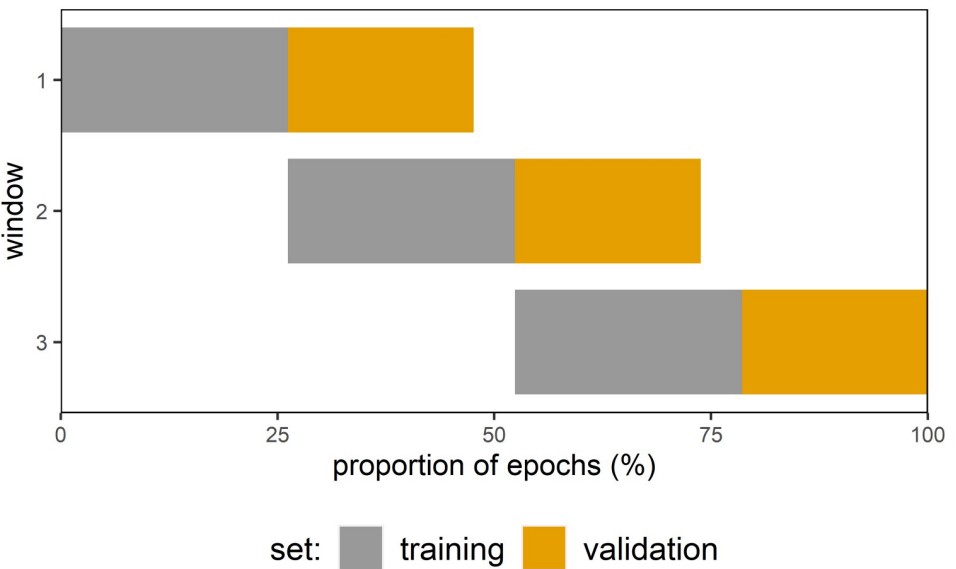

**Fig 2. Procedure of sliding window cross-validation.** For each participant, the training set was split into three consecutive windows of equal size. In each window, the model was trained with a pre-specified lambda on the initial 55% of the window's samples and then, tested on the remaining 45% samples (i.e., the validation set). After assessing the predictive performance in all windows, we finally averaged the model's performance across the windows' validation sets.

Similar to the intra-individual model evaluation, we extracted for each participant ("train participant") features using reaction time parameters and spatial filters from their training set. After individually training the EEG model with the cross-validated value for lambda, we evaluated for every other participant ("test participant") the inter-individual prediction performance (i.e., MAE) based on the samples from the test participant's hold-out set [24]. In the first iteration, we trained the EEG model with data from participant 1, the train participant, and evaluated its predictive performance 39 times (i.e., using the data from each test participant once: participant 2, 3, . . ., and 40). In the second iteration, we trained the EEG model with data from participant 2 and evaluated its performance with the data from participant 1, 3, 4, . . ., and 40. This procedure continued until we had used all participants once for training. Finally, we averaged for each train participant the MAE measured in the hold-out sets from the remaining 39 test participants.

To evaluate the inter-individual prediction performance of the EEG model, we compared for each participant the MAE of the participant's EEG model using (a) the same participant's hold-out set and (b) the hold-out sets of all other participants (whose MAE we then averaged). To inspect the impact of person-specific reaction time aggregates and spatial filters ("pre-processors") on the inter-individual predictive performance of EEG models, we evaluated the inter-individual performance additionally using the test participant's pre-processors for feature extraction.

For both the intra- and inter-individual evaluation, we finally performed paired t-tests using bootstrapping with 9999 iterations to take the skewed distribution into account and to evaluate the differences between models (a) and (b) based on the measures from all 40 participants.

## Results

Our goal was to develop individually-tailored models predicting reaction times for correct answers in a mental rotation task based on EEG features. To make generalizable predictions, we applied a machine learning approach wherein we trained person-specific linear ridge regressions. After pre-processing the EEG data and removing outliers from the training set, an average of 110 epochs (min = 76, max = 138 epochs) remained for training and an average of 40 epochs (min = 26, max = 46 epochs) for the hold-out set. For the training, we individually optimized the hyperparameter lambda through a sliding-window cross-validation procedure minimizing the prediction error (i.e., the MAE) in the validation sets. Next, we estimated the regression coefficients of the EEG features from the training set (i.e., the EEG model) to predict reaction times (S1 Fig). For the testing, we predicted reaction times using the trained model with new EEG features from the hold-out set and calculated the MAE.

### Intra-individual model evaluation

The model comparisons regarding their predictive performance revealed that the EEG model making predictions based on EEG features with a mean (M) MAE of 0.89 and a standard deviation (SD) of 0.16 performed significantly better (p < .001) than the RT model predicting the average reaction times (M = 0.91, SD = 0.16) (Fig 3). As a comparison, when we re-ran the analysis without bootstrapping, the difference remained significant (t(39) = -4.75, p < .001). When transforming the standardized and log-transformed reaction time labels back to their original scale (i.e., ms), the EEG model predicted the true reaction time labels with a mean MAE of 748 ms and the RT model with a mean MAE of 772 ms.

Given the inverse relationship between stimulus difficulty (i.e., angular disparity) and accuracy in the mental rotation task [31], the number of samples to train and test our models varied across rotational angles as we excluded samples with incorrect responses. For training, we had averaged across participants most epochs for stimuli with 0˚ rotation (M = 41), followed by 50˚ (M = 39), 100˚ (M = 37) and 150˚ rotation (M = 33). To evaluate whether the intra-individual predictive performance of the EEG model varied as a function of angular disparity, we performed a one-way repeated measures ANOVA including participant as a random effect and angle as a fixed effect nested within participant. The ANOVA revealed that there was no significant difference in the dependent variable, MAE, between the levels of the independent variable, rotational angle (F(3, 117) = 0.22, p = .88).

### Inter-individual model evaluation

Next, we evaluated the predictive performance of individually trained EEG models to predict reaction times with the last 25% of epochs from other participants (i.e., the hold-out sets). Comparing the intra-individually tested EEG model (M = 0.89, SD = 0.16) with the inter-individually tested EEG model (M = 1.59, SD = 0.67) revealed that the intra-individually tested EEG model performed significantly better (p < .001). Due to inter-individual variations in reaction times and informative oscillatory sources, it is unclear how much the drop in inter-individual prediction performance can be attributed to either the ridge regression model or to an unsuitable feature extraction.

To inspect this, we additionally evaluated the impact of individualized pre-processors (i.e., person-specific reaction time label aggregates for standardization, and spatial filters belonging to the same frequency band) on model performance. In contrast to the previous inter-individual model evaluation where we applied the train participant's pre-processors (i.e., reaction time parameters and spatial filters) to each test participant's data, we used the test participant's pre-processors for feature extraction. In both cases, we applied the train participant's ridge

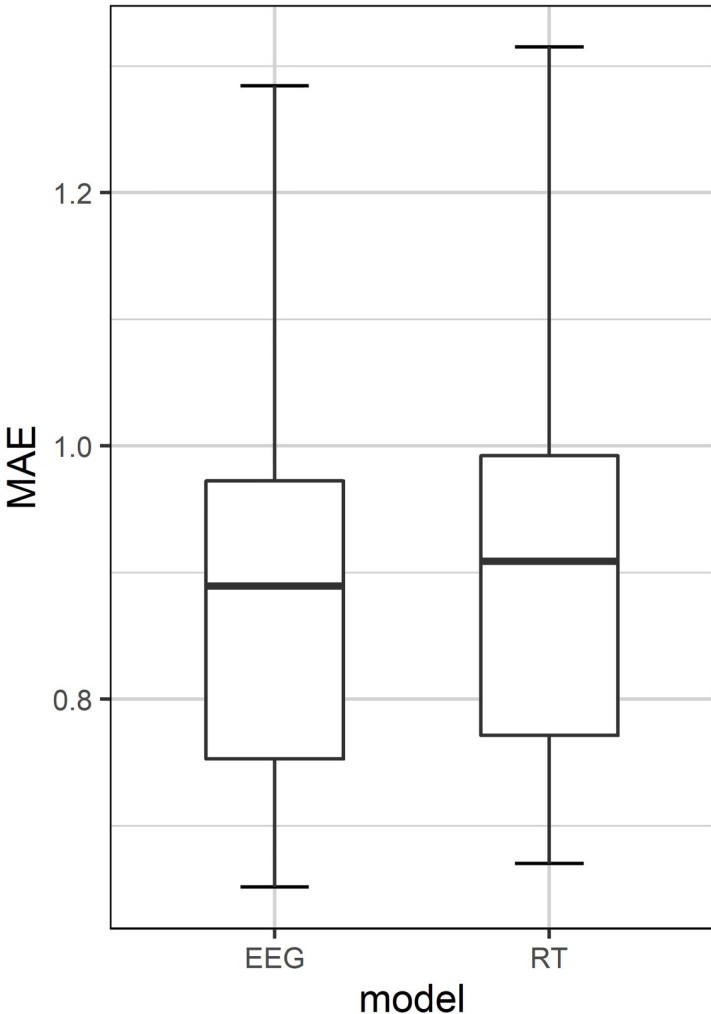

**Fig 3. Intra-individual model performance.** Boxplots of mean absolute error (MAE) in predicting standardized and log-transformed reaction times (RTs) using either a model trained on the same person's EEG features (i.e., the EEG model) or a baseline model (i.e., the RT model).

regression parameters for the prediction with the test participant's features. For final comparison, we again averaged the test participants' MAEs for each train participant. The comparison revealed that the EEG model with the test participants' pre-processors (M = 0.93, SD = 0.03) predicted the test participants' reaction times significantly better than the EEG model with the train participants' pre-processors (p < .001). We then compared the outperforming inter-individually evaluated EEG model to the intra-individually evaluated EEG model. The intra-individually evaluated EEG model slightly outperformed the inter-individually evaluated EEG model applying the test participants' pre-processors (p = .1).

## Feature importance

To assess person-specific feature importance scores of the trained ridge regression model, we relied on SHAP values. In contrast to permutation feature importance evaluating the decrease in model performance, SHAP values partition the contribution of each EEG feature to the

predicted, log-transformed and (per angle) standardized reaction time. To measure the global importance, we calculated for each participant and EEG feature the mean absolute SHAP value. In Fig 4a, we display the SHAP feature importance of a representative participant with the EEG features sorted in ascending order of frequency band. To inspect the relationship between feature values and SHAP values, we display the participant's Beeswarm plot in Fig 4b. These show that, for this participant, lower frequencies contributed more to the prediction than higher frequencies with a peak at the alpha frequency and that an increase in alpha activity was associated with a decrease in reaction time.

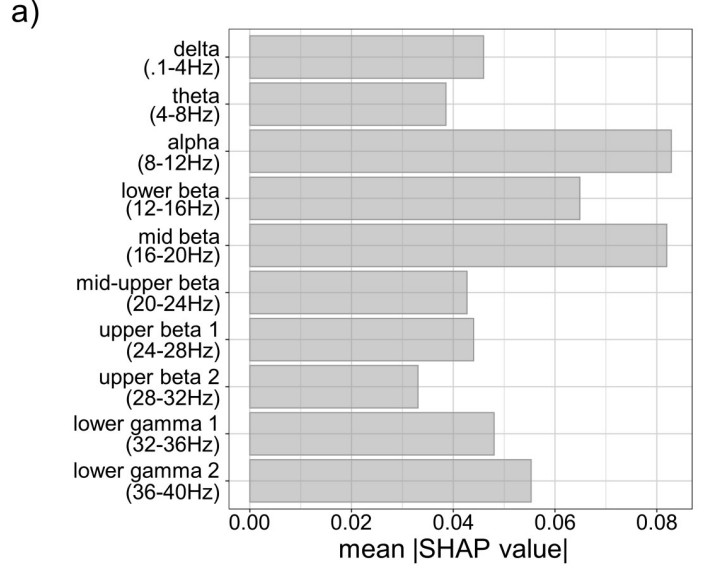

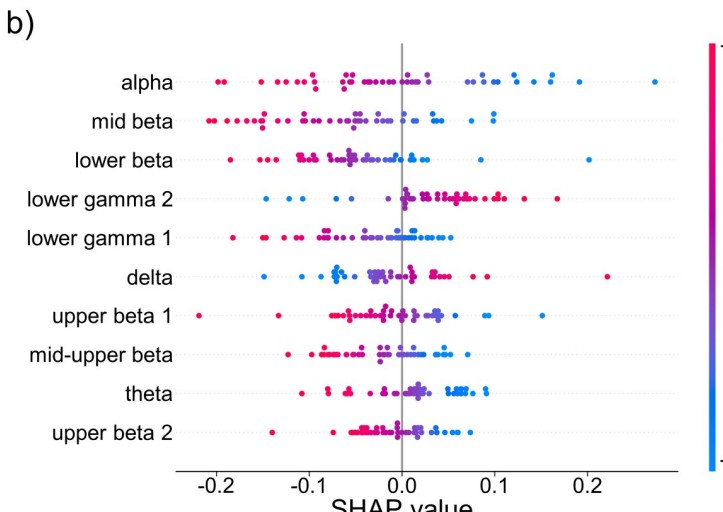

**Fig 4. Feature importance scores based on log-variance features for a representative participant.** a) Barplot of absolute SHAP values averaged per EEG feature displaying that lower frequency bands contributed more to the final prediction than higher frequencies with the alpha frequency being the most important feature. b) Beeswarm plot of SHAP values inspecting the direction of the association between EEG feature and predicted values. For this participant, the association between alpha and predicted reaction time was negative, hence, the more power in the alpha frequency band increased the more the predicted reaction time decreased.

Next, we aggregated the individual measures of feature importance across all participants to visualize patterns at the group level. In Fig 5a, we display the median of absolute SHAP values per EEG feature. These show an inverted-U shape relationship between the frequency bands and the median SHAP feature importance and, furthermore, substantial inter-individual differences in the importance of each EEG feature. Overall, frequency bands in the beta range contributed the most to the prediction of reaction times. We additionally calculated the difference in mean absolute SHAP value between all EEG features and visualized it in a heat map (Fig 5b). The pattern revealed that frequency bands closer to another differed less in their mean absolute SHAP value than more distant frequency bands.

To inspect the neural source of the SPoC components, we estimated the forward model and averaged the resulting patterns across all participants (Fig 6). Mostly, we observed an increased activity in the left frontal cortex and the right posterior parieto-occipital regions. Nevertheless, the frequency bands differed in their topographic pattern of their SPoC components correlating the band's power time course with the time course of reaction time labels. In the alpha and mid beta range activity relating to mental rotation mainly peaked in left frontal and right posterior parieto-occipital regions. In the upper beta band and lower gamma ranges the activity increased predominantly in the right occipital region. Despite the ICA routine reducing neural activity unrelated to mental rotation, we observed source activity in the frontal and lateral temporal regions of faster frequency bands typically associated with physiological artifacts.

## Discussion

The present study designed and evaluated a person-specific machine learning approach to estimate the contribution of EEG features predicting the latency of correct responses in a mental rotation task. Using established methods for modelling mental processes based on neural activity, in combination with the widely used mental rotation task capturing visuospatial performance, we successfully created person-specific models which accurately predicted reaction times based on that person's EEG activity and to a lesser degree based on another person's EEG activity. Additionally, we explored the contribution of the various EEG features to the final prediction.

First, we demonstrated that models using person-specific EEG features (i.e., EEG models) predicted the reaction times of the same person significantly more accurately than models relying on the average reaction time (i.e., RT models). Although this difference was small, the observed MAE for the EEG model are in line with previous research and represent the lower bound of what can be achieved, especially when considering the slower reaction times compared to previous studies [22, 23]. Since we excluded the reaction time of trials in which participants responded incorrectly and given the inverse relationship between angular disparity and accuracy, the number of available samples decreased with angular rotation. However, we did not find changes in predictive performance as a function of angular disparity indicating that this imbalance did not affect model performance or interpretability. These results suggest that EEG features contributed more to the prediction of reaction times than the average reaction time. Furthermore, the similarity of predictive performance across levels of rotational angles in the mental rotation stimuli suggests that the model learned from all difficulty levels (i.e., rotational angles) during training. We also found that person-specific EEG features predicted the same person's reaction times significantly more accurately than the reaction times from another person. When taking the other person's reaction time aggregates and EEG topography patterns into account, we could increase the generalizability of a person-specific EEG model. However, the person-specific EEG model continued to predict reaction times of the same person more accurately indicating that associations between EEG activity and mental rotation are

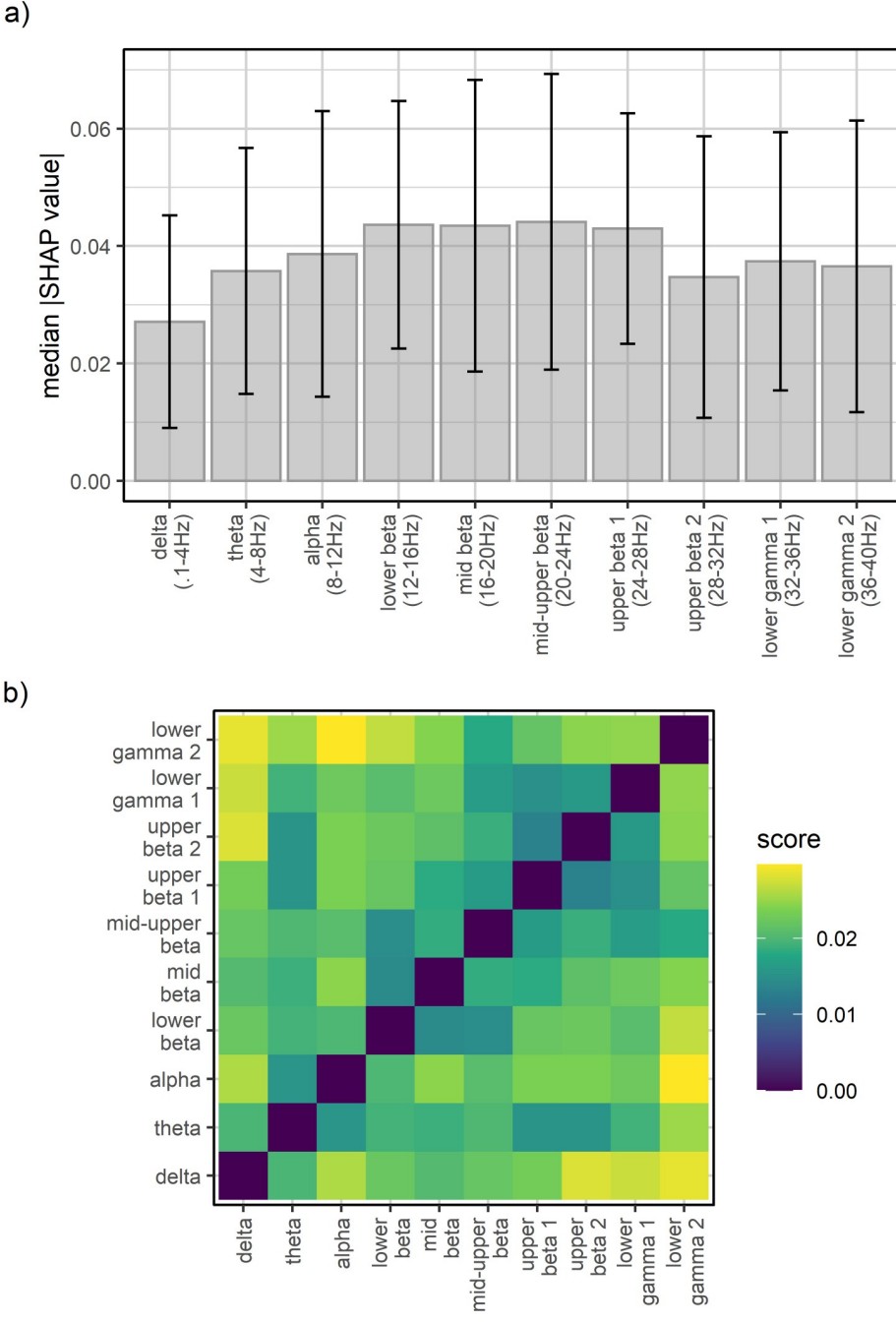

**Fig 5. Feature importance scores of all participants.** a) Barplot of the median absolute SHAP values per EEG feature with higher values for the beta frequency range indicating that these features contributed more to the final prediction than other features. Error bars denote the median absolute deviation (MAD). b) Heat map of the averaged difference in absolute SHAP value between EEG features indicates that more distant frequency bands differed more in their contribution to the prediction than frequency bands which were closer to one another.

partly specific to a person. Finally, we found that across all participants alpha and beta band related activity contributed most to the mental rotation process.

This finding is well in line with previous research, which found alpha and beta ERDs when performing a mental rotation task [6, 7]. Overall, frequency bands close to one another were

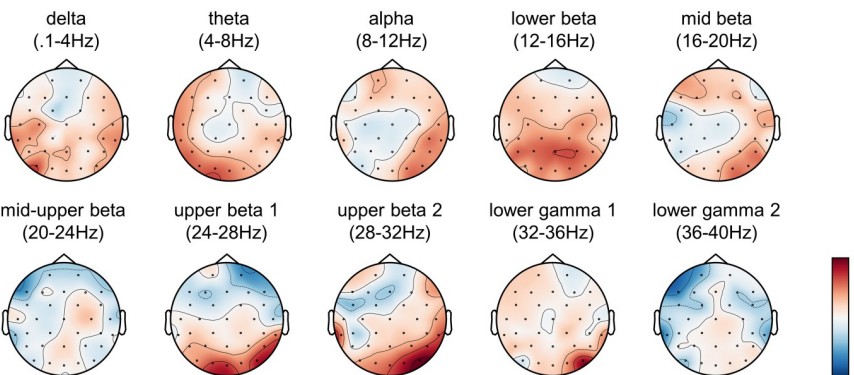

**Fig 6. Topographic patterns of average Source Power Comodulation (SPoC) components per frequency band.**
After estimating the topographic patterns using the forward model for each participant, we averaged the activation pattern across all participants per frequency band.

more similar regarding their importance for the prediction of reaction times than distant bands (Fig 5b) which supports previous work based on highly similar spatiotemporal characteristics between sub-bands [43]. Further, we also measured increased activity in left frontal and right parieto-occipital regions during mental rotation typically observed in mental rotation of non-bodily stimuli [1, 2]. It is noteworthy that these studies focused on the association between frequency band activity and mental rotation task performance in the general population and did not investigate the predictive performance on unseen data. Hence, our results extend these findings by demonstrating similar frequency bands minimizing the prediction error using a machine learning approach. Furthermore, the results of the present study provide evidence for inter-individual variations of the associations between frequency bands and mental rotation performance. Taking individual topographical patterns into account was of major importance in generalizing one person's model to the data from another person. However, the EEG model continued to predict reaction times more accurately for the same person it was trained on than for another person suggesting that leave-one-participant-out analyses may fail to uncover topographical patterns beneficial for within-person predictions. This is in line with individually trained classifiers to detect task states based on neural activity which performed well above chance for the same person and (to a lesser degree) for others [24].

Overall, our results indicate that a more individualized approach towards modelling neural activity and behavioural variables may be a promising approach for neuropsychological research and clinical applications, particularly when combined with machine learning. This may enable person-specific adjustments of interventions targeting neural activity to individually maximize the effect on behavioural outcomes. Our findings also suggest that individual variations in behaviour and topographical activity patterns have a large effect on the generalizability of a model trained on one person's data. To understand which neural activity pattern the brain relies on when processing visuospatial information, we proposed the use of SHAP values to estimate the contribution of neural activity patterns to behavioural responses. Although this should be seen as early evidence, the pattern of SHAP values across participants–that is, alpha and beta band related activity contributed the most to the final prediction–replicates previous findings [6, 7]. Furthermore, there was significant inter-individual variability in the importance of neural activity patterns for processing visuospatial information suggesting that the reliance on specific frequency bands in brain networks may be partly person-specific. Notably, these results are based on a small sample size (i.e., between 105 and 186

samples per person) indicating that our estimates represent a lower bound of what can be achieved.

Despite these promising results and their potential implications, it is important to point out some of the major limitations. For data collection we administered a mental rotation task in which the duration of each trial was determined either by the participant's response or a time-out after 7.5 seconds. While this procedure enabled us to capture neural activity related to mental rotation (in contrast to other implementations in which the presentation duration of the 3D objects is fixed in advance), the SPoC algorithm required epochs of equal size as input. Although our model performed similarly in the prediction of reaction times across varying degrees of angular rotation indicating that the final epochs contained information of multiple stages of mental rotation, meaningful information may have been lost when cutting the varying epoch sizes into a standard size. The relatively small number of epochs used to train person-specific regression models predicting reaction time labels with EEG features risks yielding non-robust regression parameters (e.g., rank switches in feature importance estimates). To estimate more robust spatial filters, a regularized SPoC algorithm represents a promising approach for future studies [44]. While we expected to reduce within-person variation of reaction times unrelated to mental rotation by instructing participants to choose a response strategy, this may have resulted in inter-individual differences of the response strategy, partly impacting the generalizability of person-specific EEG models. Furthermore, we would like to point out that the used data were collected in a single session and that we neither assessed the congruent validity of our models with other tasks measuring visuospatial ability nor the discriminant validity regarding processes unrelated to mental rotation. To tune the shrinkage parameter, we applied a three-fold sliding-window cross-validation procedure which worked well for our data and generated the expected U-shaped curve representing the bias-variance trade-off. However, we did not assess how our hyperparameter optimization procedure compares to other methods for defining the shrinkage parameter which may outperform our routine. Finally, since we collected observational data of neural activity patterns and behavioural responses, the models should be interpreted with caution and do not imply causality.

Further research is required to overcome these limitations and to extend our knowledge about the relationship between neural activity and behavioural variables. Our results indicate that an individualized machine learning approach towards modelling behaviour with neural activity could be promising for neuropsychological research. This may be particularly powerful for changing behaviour based on modulations of neural activity. To test for causality, one approach would be to experimentally manipulate EEG features contributing the most to the prediction and to measure the target behaviour before and after the manipulation (e.g., with neurofeedback, transcranial magnetic stimulation, or transcranial direct current stimulation). Future research should also investigate the reliability and validity of these individualized models. Further, the design of an appropriate task and methodology to capture meaningful neural activity during mental rotation requires further attention. We artificially constrained the duration of recorded neural activity preceding the behavioural response to apply the SPoC algorithm for further processing. One interesting approach would be to investigate how additional time for mental rotation would improve modelling the association with neural activity by pre-specifying the presentation duration of the 3D objects during a mental rotation task in incremental steps. As artificially delaying a response may invoke processes unrelated to the decision [45], novel algorithms processing EEG signals which deal with varying epoch sizes may be as beneficial. Finally, comparing different time windows (e.g., initial 500 vs last 500 ms) informed by EEG microstate analysis may reveal how spectral and spatial EEG features differ between stages of cognitive processing (e.g., encoding of visual information, mental rotation, and decision making) [39].

To conclude, we present an individually-tailored machine learning approach to model mental rotation as a function of neural activity. Therefore, we trained ridge regression models to predict reaction times in trials of a mental rotation task where participants responded correctly using EEG features which take individual topographical activity patterns into account. To this end, we showed that individualized models generate more precise predictions than when relying on a model from another person. Furthermore, neural activity preceding the response in the mental rotation task predicted the time of the response significantly more accurately than predictions relying on the average reaction time of that person. Finally, we demonstrated that taking individual variations in response patterns and topographical activity patterns into consideration significantly improves the generalizability of a model trained on one person's data to data from another person. Given the observational nature of the data used, further research is required to establish the causal effect of specific frequency bands on the mental rotation performance. Nonetheless, our research represents an early step towards individualized neurocognitive models and, finally, towards highly specified treatment options.

## Supporting information

**S1 Fig. Reaction time histograms.** Histograms visualizing the distribution of participant-level reaction time (RT) averages per angular rotation (i.e., 0˚, 50˚, 100˚, 150˚) and unit (i.e., ms, log (ms)) included as labels in the training set. The dotted lines indicate the average RT per angular rotation and unit.
(TIF)

## Acknowledgments

The authors are grateful to Prof. André Schulz (Department of Behavioural and Cognitive Sciences, University of Luxembourg) for providing access to the CLIPSLAB facilities to carry out the data collection. The authors are also grateful to Dr. Annika Lutz (Department of Behavioural and Cognitive Sciences, University of Luxembourg) and Mr. Karsten Schönbein (MediaCentre, University of Luxembourg) for their technical support on the EEG system used.

## Author Contributions

**Conceptualization:** Sinan Uslu.

**Data curation:** Sinan Uslu.

**Formal analysis:** Sinan Uslu.

**Funding acquisition:** Claus Vögele.

**Investigation:** Sinan Uslu.

**Methodology:** Sinan Uslu.

**Project administration:** Claus Vögele.

**Software:** Sinan Uslu.

**Supervision:** Michael Tangermann.

**Visualization:** Sinan Uslu.

**Writing – original draft:** Sinan Uslu.

**Writing – review & editing:** Michael Tangermann, Claus Vögele.

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
