## [Decision Letter · Decision Letter 0]

20 Jul 2023

PONE-D-23-18489Estimating person-specific neural correlates of mental rotation: A machine learning approachPLOS ONE

Dear Dr. Uslu,

Thank you for submitting your manuscript to PLOS ONE. After careful consideration, we feel that it has merit but does not fully meet PLOS ONE’s publication criteria as it currently stands. Therefore, we invite you to submit a revised version of the manuscript that addresses the points raised during the review process.

We look forward to receiving your revised manuscript.

Kind regards,

Humaira Nisar

Academic Editor

PLOS ONE

We will update your Data Availability statement on your behalf to reflect the information you provide."

3. We notice that your supplementary figure is uploaded with the file type 'Figure'. Please amend the file type to 'Supporting Information'. Please ensure that each Supporting Information file has a legend listed in the manuscript after the references list.

Additional Editor Comments:

Please respond to all comments by the reviewers.

Reviewers' comments:

Reviewer's Responses to Questions

**Comments to the Author**

1. Is the manuscript technically sound, and do the data support the conclusions?

Reviewer #1: Partly

Reviewer #2: Yes

Reviewer #3: Yes

2. Has the statistical analysis been performed appropriately and rigorously? 

Reviewer #1: N/A

Reviewer #2: I Don't Know

Reviewer #3: Yes

3. Have the authors made all data underlying the findings in their manuscript fully available?

Reviewer #1: No

Reviewer #2: No

Reviewer #3: No

4. Is the manuscript presented in an intelligible fashion and written in standard English?

Reviewer #1: Yes

Reviewer #2: Yes

Reviewer #3: Yes

5. Review Comments to the Author

Reviewer #1: I have a general comment and some specific remarks about the work.

General comment.

From what I understand, the authors proposed a ML method to extract reaction times (RT) from EEG data during a mental rotation task. The “standard” method for evaluating RT from EEG data is to estimate the latencies of the components of the evoked potentials from the individual EEG. Can the authors make a comment on that?

I report my remarks (divided into Sections) in the followings:

1) Abstract

Please, add a sentence in the Abstract (and then in the Introduction) to describe mental rotation task, e.g., “recognizing what an object may look like when viewed from other angles or when oriented differently in space…”

Lines 13-14. Please, justify how mental rotation task is used to monitor progressive neurological disorders.

Line 19. Please substitute non-invasive electrocortical activity with Electroencephalographic (EEG) activity

2) Introduction

Lines 54-67. Please, describe quantitatively the meaning of “reduced” mental rotation performance (Is it quantified as greater reaction times? Greater errors?...)

Line 83. Please, define the abbreviation ERD

3) Materials and Methods

Line 126. Participants. Please, specify also the % of male and female and age distribution.

Lines 163-196. Data analysis.

Please, report formulas of the ridge regression model, describing its parameters.

Please, list the EEG features that were extracted and exploited in the model.

Details in the pre-processing are missing. Please, specify how and which artifacts were removed (if data were discarded due to artifacts, specify percentage), kind and order of filters, percentage of epochs removed per subject during pre-processing.

Lines 198-206. Please, specify the final values of lambda.

Lines 248-249. Please, describe quantitatively the “SPoCλ algorithm [24] as implemented in the MNE framework (version 0.23.0)”

4) Discussion

Lines 411-412. The authors stated: “The present study designed and evaluated a person-specific machine learning approach to extract neural correlates of mental rotation.” Are the neural correlates the reaction times? Please avoid these generic expressions and be accurate in your choice of words.

Reviewer #2: The authors present an individually tailored prediction model (ridge regression) based on EEG features to predict mental rotation reaction times (for correct answers only). They compare it to another prediction model based on task difficulty (angular disparity of rotating 3D objects) which also predicts reaction time. However authors repeatedly write as if they are basing the second model on the average reaction time to predict reaction time, which brings confusion to what are the authors exactly using as input data here (angular disparity or something else?) For instance, one would assume only from reading the abstract that this average reaction time is related to the average RT of all subjects.

The paper is written in proper English, however it is long to read as it has many repetitions and a lack of examples. It seems that the authors give many details on results that are not essential to the final findings. For the sake of readability, I would suggest to shorten some technical details and results (and their repetitions) that do not bring clarity to the paper.

The authors have used an interesting method to extract EEG features, using SpoC spatial filter. The methods for EEG signal processing and artifact removal seem correct, The experiment itself is also well designed, LSL was wisely used, with a baseline (idling period) and mental rotation task, with enough repetitions per participant. Also, the limitations are clearly stated in the discussion...

My major concern is that the authors do not explain the motivation for using several methods. For this reason, I am not sure of their validity either.

In the Hyperparameter Tuning. I fail to understand the motivation behind the choice of the quantity of windows (why precisely 3), why are the windows overlapping, and why choose 55% for training and 45% for validation set of the model.

Also Figure 1 does not bring any clarity to the method used.

Most of all, why are the authors using this chronological cross validation that overlaps? Please elaborate.

In the Feature Extraction. The authors should give detailed explanation to why they perform such transformations of reaction times. Also reformulate the text to highlight that we are looking at X as EEG data (bank filtered frequencies), and that the labels which represent the reaction times should be normalized in order to be in the same scale as the log values of EEG. Please add a reminder here of the minimum and maximum duration of RT (to have an idea of the scaling or normalization per object angles).

The scaling of the reactions times per angular rotation makes sense, and the one example nicely clarifies this. However, the figure 2 is saturated with information. Please write X as EEG data, Y as RT, remind what is the value 10 (in Nx10) etc.

The equation of the prediction model is placed too soon in the fig 2, it should be written as a separate equation or in another figure later on. The fig 2 should instead be depicting just the transformation of reaction times and EEG freqs, and the standardization of data, in order to make the reasoning behind it more comprehensible.

Precise please that the log transformation of reaction time is done per subject and not for all participants; please give clear descriptions in this section. Also, how was 1.5 chosen as threshold for MAD?

In Fig 2 I find that the model evaluation is placed too soon in the figure, there should be a separate one relating to it; especially as the authors are mentioning spatial filtering before explaining the procedure.

Line 254 Please give an example how the univariate signal was made. This whole section could benefit from more examples. Is there one value (feature) per band or one value for all bands (it is clearer when looking at the results, but it should also be clear here). Remind how many frequency bands there are, and please give an example of a feature,

The Result section contains explanations of the methods which bring confusion. This section should only contain results and not repetition of methods (or at least shorten the reminders).

Line 309, it is unnecessary for a reader to know both values of reaction time, just use the one in ms. Details such as this one make the paper difficult to read. Please choose the information such that is essential for the paper findings.

For ANOVA, please write dependent and independent variables, and if it is 1-way or not... Was an Anova performed for EEG model as well? Here, there is only the RT model?

Clarify whether the RT model is intra-individual as well, and what does the average RT represent exactly (avg per how many epochs, give example).

In the result section, the authors explain inter-individual method that should be explained in the methods section, before the results, and with examples.

Line 334 might be an error, as there should be 39 train participants for the one remaining to be tested? It seems like there is an overfitting here. Again, not very clear without examples. Do the authors perform LOSO (leave one subject out) evaluation?

Here, the authors also mention individualized "pre-processors" which were not mentioned earlier. I suggest to either delete this part or make it more clear (and write it in the right section), with examples. Please also write the motivation behind this comparison, and maybe choose one meaningful result.

In the Discussion there are again explanations of the methods; It is fair to give a few short reminders but there is no need for detailed and repeated explanations that should be stated in the methods sections.

Line 442, the authors write about relying on the average RT again which remains confusing. How are average RTs predicting RTs?

The limitations are nicely written, however the Discussion is again very long and it seems repetitive with the rest of the paper; please shorten the paper overall.

The paper simply needs re-writing but seems to be correct; I shall know better when it is more understandable. The only true issue is this sliding cross validation method for lambda tuning; I am not sure I understand why this method was chosen, and if it is completely valid...

Reviewer #3: Overview

This manuscript reports a compelling investigation into the potential of an individually tailored machine learning approach to identify EEG patterns of neural activity during mental rotation. The introduction provides extensive background on clinical groups associated with changes in the performance of mental rotation tasks. The final section presents the ultimate goal of this research as “the identification of the neural substrates may represent a promising approach for neurophysiological stimulation studies to finally restore impaired behavioural functionality.”

The proposed machine learning model is based on spectral EEG features (10 frequency bands) and the target is based on the reaction times of the correct answers. The definition of the target vector takes into account the specificity of the task (rotation angle and reaction time). The hyperparameter of the model is optimized in a cross validation framework. The trained model is tested in an hold-out set and intra-individual prediction performance assessed using Mean Absolute Error. Feature importance is analyzed based on the SHapley Additive exPlanations (SHAP) method. The authors evaluated two models: the EEG model (intra-individual), which assessed individual prediction performance, and the RT model, which predicted the average, standardized reaction time per rotation angle (group-level).

The paper's main contribution is a significant difference between intra-individual model and an inter-individual evaluation regarding the prediction of the reaction time of a mental rotation task.

The paper is well-written, but may improve its comprehensibility. The methods are straightforward but rather puzzling in some aspects. The results are well-framed in the literature and pave the way for future studies.

Several aspects caught my attention, and I would like the authors to address and discuss them in a revised version.

Comments

Why not instruct the participant to follow the same strategy (response as soon as possible)? (“We instructed participants to choose a strategy for responding (i.e., either slower and more accurate or faster and less accurate) and to stick to it throughout the task.”) This is critical since one of the variables of interest is the reaction time. Is there a relation between the number of epochs available after preprocessing and the strategy followed by each participant?

It is not clear when was the window of 500ms selected - after the trigger corresponding to the presentation of the two images? why not using 500ms before the button press preparation (from -700ms to -200ms)? This would correspond to the final stage of decision making. Comparing the classification results between both windows would also inform regarding the importance of both stages.

The authors present the hyperparameter tuning after epoch extraction. However, I believe that it would be easier to understand after feature extraction, as these feature sets will be used in model training during the cross-validation procedure.

The definition of two evaluation targets (EEG model and RT model) should be addressed earlier as well as the objectives of each model. The EEG model is very nicely explained. However, the RT model is quite puzzling to me - I suggest the authors to describe it independently as it is not clear to me which was the training and the target (as far as I understand, it represents a group analysis - the same features and target - instead of an individual one, i.e., the training of the RT model includes all participants?).

Line 306: Considering the results of the intra-individual and RT model, it is rather strange that the results and significant - add statistical details of the test.

One alternative to inter-individual model evaluation would be to bootstrap across participants, i.e. train the model with 39 part and test in the remaining one (repeating this process for all participants). What is the comment of the authors regarding this hypothesis?

The model presented in line 344 (second approach to inter-individual evaluation) is quite puzzling. As far as I understand, the features used to train the model and to test it are not the same (different participants may have different parameters).

Minor comments

page 5, line 84: Additional information on the features considered by the classifiers. Additionally, please re-phrase “achieved a mean absolute error between 100 and 600ms”

line 190: “an insufficient amount of data was recorded”. This happened due to an early response (<700ms) of the participant? please address this in the manuscript.

The presentation of the SHAP values for a representative participant (e.g. figure 4) is not necessary in my opinion and could be presented in suppl. materials. Figure 5b is also quite strange as no discussion/interpretation is presented on this results.

6. PLOS authors have the option to publish the peer review history of their article (what does this mean?). If published, this will include your full peer review and any attached files.

Reviewer #1: No

Reviewer #2: No

Reviewer #3: **Yes: **Bruno Direito

---

## [Author Response · Author response to Decision Letter 0]

20 Sep 2023

Note:

While line numbers in the comment section refer to the original manuscript, line numbers in the response section refer to the revised version. As we changed the order of Fig 1 and 2, they also switched their number (in the response section). Additionally, we have corrected two typos in the revised manuscript (lines 219 and 229): “500 x 32 samples” (instead of 5000 x 32 samples).

Comments by the journal:

Comment 1: Please ensure that your manuscript meets PLOS ONE's style requirements, including those for file naming. The PLOS ONE style templates can be found at https://journals.plos.org/plosone/s/file?id=wjVg/PLOSOne_formatting_sample_main_body.pdf and https://journals.plos.org/plosone/s/file?id=ba62/PLOSOne_formatting_sample_title_authors_affiliations.pdf

Response 1: We are grateful for the reminder and ensured that the applied style meets the requirements.

Comment 2: We note that you have indicated that data from this study are available upon request. PLOS only allows data to be available upon request if there are legal or ethical restrictions on sharing data publicly. For more information on unacceptable data access restrictions, please see http://journals.plos.org/plosone/s/data-availability#loc-unacceptable-data-access-restrictions.

We will update your Data Availability statement on your behalf to reflect the information you provide."

Response 2: In our revised cover letter, we now explain that we cannot share the data due to ethical and legal restrictions, and added the required information: “The data contain sensitive information and cannot be shared publicly due to GDPR. Access will require approval from the Ethics Review Panel (erp-submissions@uni.lu) and the Data Protection Office (dpo@uni.lu) of the University of Luxembourg.”

Comment 3 We notice that your supplementary figure is uploaded with the file type 'Figure'. Please amend the file type to 'Supporting Information'. Please ensure that each Supporting Information file has a legend listed in the manuscript after the references list. 

Response 3: We have amended the file type.

Comments by reviewer 1

Comment 1 I have a general comment and some specific remarks about the work.

General comment.

From what I understand, the authors proposed a ML method to extract reaction times (RT) from EEG data during a mental rotation task. The “standard” method for evaluating RT from EEG data is to estimate the latencies of the components of the evoked potentials from the individual EEG. Can the authors make a comment on that? 

Response 1 We thank the reviewer for their comments. We decided to use the behavioural reaction time as this outcome has been used in other machine learning approaches*. Furthermore, we see neurofeedback as a potential application domain of our approach and neurofeedback commonly targets behavioural changes.

* Binias B, Myszor D, Palus H, Cyran KA. Prediction of pilot’s reaction time based on EEG signals. Front Neuroinform. 2020;14(6). doi:10.3389/fninf.2020.00006

* Rahman SU, O’Connor N, Lemley J, Healy G. Using pre-stimulus EEG to predict driver reaction time to road events. Annu Int Conf IEEE Eng Med Biol Soc. 2022:4036–4039. doi:10.1109/EMBC48229.2022.9870904

Comment 2: Please, add a sentence in the Abstract (and then in the Introduction) to describe mental rotation task, e.g., “recognizing what an object may look like when viewed from other angles or when oriented differently in space…”

Response 2: We have added a short description of the mental rotation task to the abstract (lines 13-14): “The mental rotation task requires objects to be mentally rotated in space. It has frequently been used to monitor progressive neurological disorders.”

Comment 3: Lines 13-14. Please, justify how mental rotation task is used to monitor progressive neurological disorders.

Response 3: We have modified the sentence (lines 13-14): “The mental rotation task requires objects to be mentally rotated in space. It has been used to monitor progressive neurological disorders.” and added a justification (lines 53-56): “In patients showing progressive behavioural and neurological decline, as for example in those with Huntington’s disease, subtle impairments in visuospatial abilities found in early stages transition to significant differences in symptomatic stages when compared to healthy controls [8].”

8. Glikmann-Johnston Y, Fink KD, Deng P, Torrest A, Stout JC. Spatial memory in Huntington’s disease: A comparative review of human and animal data. Neuroscience & Biobehavioral Reviews. 2019;98: 194–207. doi:10.1016/j.neubiorev.2019.01.015

Comment 4: Line 19. Please substitute non-invasive electrocortical activity with Electroencephalographic (EEG) activity

Response 4: We have adapted the sentence accordingly (lines 19-20): “[…] using task-related, electroencephalographic (EEG) activity of the same person.”

Comment 5: Lines 54-67. Please, describe quantitatively the meaning of “reduced” mental rotation performance (Is it quantified as greater reaction times? Greater errors?...)

Response 5: We have specified the performance measures (lines 59-60): “[…] showed reduced mental rotation performance manifesting in both higher reaction times and error rates [9]”

Comment 6: Line 83. Please, define the abbreviation ERD

Response 6: The term is defined at its first appearance in lines 49-50.

Comment 7: Line 126. Participants. Please, specify also the % of male and female and age distribution.

Response 7: We have added the requested information (lines 126-129): “We collected the data in the context of a larger study for which we invited 40 participants (25 females, mean age: 24.97 years, age range: 19-35 years) to single two-hour laboratory assessments at the University of Luxembourg between September and November 2022.”

Comment 8: Lines 163-196. Data analysis.

a) Please, report formulas of the ridge regression model, describing its parameters.

b) Please, list the EEG features that were extracted and exploited in the model.

c) Details in the pre-processing are missing. Please, specify how and which artifacts were removed (if data were discarded due to artifacts, specify percentage), kind and order of filters, percentage of epochs removed per subject during pre-processing.

Response 8: a) we have added a reference to figure 2 as we describe the model in the figure caption (lines 168-170): “We trained a ridge regression model to predict the reaction time of correct responses in the mental rotation task based on features extracted from the EEG data prior to the response (Fig 1).”

b) we have added a reference to the section where we detail the feature extraction procedure (lines 175-176): “For details on the EEG features used to train our models, see the subsection on feature extraction.”

c) we have added a reference to the subsection where we provide more details (lines 170-173): “First, we pre-processed the EEG data which involved removal of artifacts, bandpass filtering, the epoching of the continuous EEG signal and the removal of epochs with a duration of less than 700 ms (for more details, see the subsection on pre-processing).” There, we have added more details on the filters (lines 202-205): “To prepare the extraction of band power features from the cleaned EEG signals, we applied ten non-overlapping, consecutive frequency filters (width: 4 Hz)one-pass, non-causal, zero-phase, Hamming-windowed finite impulse filters that extracted the central frequencies at 2, 6, 10, 14, 18, 22, 26, 30, and 34 Hz with a width of 4 Hz [26,38].” and the percentage of epochs removed per participant (lines 216-219): “After removing on average 15.3 % of epochs with 3D figures presented due to missing or too fast responses, this procedure yielded on average across all participants 163 epochs (min = 105, max = 186 epochs) with 500 x 32 samples per epoch.”

Comment 9: Lines 198-206. Please, specify the final values of lambda.

Response 9: We have added the median, minimum, and maximum value for lambda (lines 267-270): “For each lambda we stored the average of all three windows’ MAE. Finally, the value for lambda with the lowest average MAE score, which was on median 241.13 (min = 3.53, max = 2476.37) , was chosen for further processing.”

Comment 10: Lines 248-249. Please, describe quantitatively the “SPoCλ algorithm [24] as implemented in the MNE framework (version 0.23.0)”

Response 10: We have added a reference to the respective documentation which gives detailed information on the algorithm and provides a link to the source code (lines 248-251): “More specifically, we applied the SPoCλ algorithm [24] as implemented in the MNE framework (version 0.23.0), which maximizes the covariance between the two variables of interest (more details can be found in the documentation [41]).”

Comment 11: Lines 411-412. The authors stated: “The present study designed and evaluated a person-specific machine learning approach to extract neural correlates of mental rotation.” Are the neural correlates the reaction times? Please avoid these generic expressions and be accurate in your choice of words.

Response 11: We have modified our text and specified the term neural correlates (lines 438-440): “The present study designed and evaluated a person-specific machine learning approach to estimate the contribution of EEG features predicting the latency of correct responses in a mental rotation task.”

Comments by reviewer 2

Comment 1: The authors present an individually tailored prediction model (ridge regression) based on EEG features to predict mental rotation reaction times (for correct answers only). They compare it to another prediction model based on task difficulty (angular disparity of rotating 3D objects) which also predicts reaction time. However authors repeatedly write as if they are basing the second model on the average reaction time to predict reaction time, which brings confusion to what are the authors exactly using as input data here (angular disparity or something else?) For instance, one would assume only from reading the abstract that this average reaction time is related to the average RT of all subjects.

Response 1: We have used for each participant the average of past reaction times per angle to predict future reaction times

Comment 2: The paper is written in proper English, however it is long to read as it has many repetitions and a lack of examples. It seems that the authors give many details on results that are not essential to the final findings. For the sake of readability, I would suggest to shorten some technical details and results (and their repetitions) that do not bring clarity to the paper.

Response 2: We thank the reviewer for these suggestions. We have removed the repetitions (see our response to comments #8, 11, 13, 15) and added some examples (see our response to comments #7, 11). Furthermore, we have shortened the results section and its repetition in the discussion (see our response to comments #8, 11, 13, 15).

Comment 3: The authors have used an interesting method to extract EEG features, using SpoC spatial filter. The methods for EEG signal processing and artifact removal seem correct, The experiment itself is also well designed, LSL was wisely used, with a baseline (idling period) and mental rotation task, with enough repetitions per participant. Also, the limitations are clearly stated in the discussion...

My major concern is that the authors do not explain the motivation for using several methods. For this reason, I am not sure of their validity either.

Response 3: We thank the reviewer for the major concern and look forward to clarifying the motivation for the methods used in the following.

Comment 4: In the Hyperparameter Tuning. I fail to understand the motivation behind the choice of the quantity of windows (why precisely 3), why are the windows overlapping, and why choose 55% for training and 45% for validation set of the model.

Also Figure 1 does not bring any clarity to the method used.

Most of all, why are the authors using this chronological cross validation that overlaps? Please elaborate.

Response 4: The choice of window size is based on exploration and has worked well for the minimization of MAE for different lambdas. To clarify, we added the following statements to the limitations section (lines 523-528): “To tune the shrinkage parameter, we applied a three-fold sliding-window cross-validation procedure which worked well for our data and generated the expected U-shaped curve representing the bias-variance trade-off. However, we did not assess how our hyperparameter optimization procedure compares to other methods for defining the shrinkage parameter which may outperform our routine.”

To explain the choice of chronological cross-validation and overlapping windows, we have added the following statements to the section on ‘Hyperparameter tuning’ (lines 259-264): “We applied a chronological cross-validation to take nonstationarities of the recorded EEG data into account [42] and defined overlapping windows to use most of the limited sample size for training. Importantly while we re-used some of the preceding window’s validation set to train the consecutive window’s train set, we did not re-use any sample for validation to reduce the risk of leakage (Fig 2).”

42. Lemm S, Blankertz B, Dickhaus T, Müller K-R. Introduction to machine learning for brain imaging. NeuroImage. 2011;56: 387–399. doi:10.1016/j.neuroimage.2010.11.004

Comment 5: In the Feature Extraction. The authors should give detailed explanation to why they perform such transformations of reaction times. Also reformulate the text to highlight that we are looking at X as EEG data (bank filtered frequencies), and that the labels which represent the reaction times should be normalized in order to be in the same scale as the log values of EEG. Please add a reminder here of the minimum and maximum duration of RT (to have an idea of the scaling or normalization per object angles).

Response 5: To explain why we applied those transformations, we have added the following statements (line 221-225): “To transform the recorded data into the same scale and to increase the signal to noise ratio, we applied a series of transformations to both the reaction time labels and the EEG signals to finally extract the EEG features based on bandpass filtered EEG signals (Fig 1). Due to their positively skewed distribution, we log-transformed the reaction times to approximate a Gaussian distribution.”

As we replaced in Fig 1 X with EEG data (see our response to comment #6), we did not further highlight that we are looking at X as EEG data.

We have added a reference to S1 Fig (lines 235-237): “To take the varying difficulty levels based on angular disparity into account, we standardized the log-transformed reaction times per angular rotation (S1 Fig).”

Comment 6: The scaling of the reactions times per angular rotation makes sense, and the one example nicely clarifies this. However, the figure 2 is saturated with information. Please write X as EEG data, Y as RT, remind what is the value 10 (in Nx10) etc.

The equation of the prediction model is placed too soon in the fig 2, it should be written as a separate equation or in another figure later on. The fig 2 should instead be depicting just the transformation of reaction times and EEG freqs, and the standardization of data, in order to make the reasoning behind it more comprehensible.

Precise please that the log transformation of reaction time is done per subject and not for all participants; please give clear descriptions in this section. Also, how was 1.5 chosen as threshold for MAD?

In Fig 2 I find that the model evaluation is placed too soon in the figure, there should be a separate one relating to it; especially as the authors are mentioning spatial filtering before explaining the procedure.

Response 6: To specify the origin of the value 10, we have modified the caption of Fig 1 (line 186): “[…] with ten EEG features […]”.

In figure 1, we have replaced X with EEG data and y with RT. We have decided to completely remove the equations from the manuscript because the equations are well-known from the literature. Hence, we have updated the caption of Fig 1 (lines 180-191): “Fig 1. Feature extraction and model evaluation. For each participant, the first 75% of epochs were used as a training set and the remaining 25% as a hold-out set. In the training set, we first removed epochs which we classified as outliers (not visualized in the figure) and then extracted the EEG features by first bandpass and then spatially filtering the EEG signal and the labels by standardizing the log-transformed reaction times. We trained a ridge regression model to minimize for all epochs (N) the sum of squared errors between the reaction time labels and the model predictions with ten EEG features while penalizing the sum of squared coefficients with the optimized penalty term lambda. For the evaluation, we applied the same spatial and frequency filters and label standardization procedure to the hold-out set to extract new features and used the trained model to generate new predictions whose error we finally measured with the recorded reaction times in the hold-out set (see filled arrow in the figure).”

Furthermore, we thank the reviewer for spotting the typo regarding the MAD threshold. It should be 2.5 x MAD and is based on a general recommendation. We have corrected the statement (lines 225-228): “Exclusively in the training set we then calculated the median absolute deviation of log-transformed reaction time labels per angle (i.e., 0°, 50°, 100°, 150°) and removed on average 12 epochs which we considered as outliers (i.e., if the label was beyond a threshold of 21.5 x MAD [40])

40. Leys C, Ley C, Klein O, Bernard P, Licata L. Detecting outliers: Do not use standard deviation around the mean, use absolute deviation around the median. Journal of Experimental Social Psychology. 2013;49: 764–766. doi:10.1016/j.jesp.2013.03.013

We have updated the text to specify that the transformations were applied for participants separately (lines 235-237): “To take the varying difficulty levels based on angular disparity into account, we standardized per participant the log-transformed reaction times for each angular rotation (S1 Fig).”

Comment 7: Line 254 Please give an example how the univariate signal was made. This whole section could benefit from more examples. Is there one value (feature) per band or one value for all bands (it is clearer when looking at the results, but it should also be clear here). Remind how many frequency bands there are, and please give an example of a feature,

Response 7: We have added the following example (lines 240-245): “For the EEG features, we started with the ten bandpass filtered signals each of which consisted of 32 channels. For each frequency band (e.g., alpha), we then applied a spatial filter transforming the 32-channel signal into a univariate time series to increase the signal to noise ratio. Finally, we performed per frequency band and epoch (i.e., 500 EEG time points) a log-variance transformation of the univariate signal to approximate the band power, the final EEG feature, which yielded ten values per sample (Fig 1).”

Comment 8: The Result section contains explanations of the methods which bring confusion. This section should only contain results and not repetition of methods (or at least shorten the reminders).

Response 8: We have removed the methods explanations from the Results sections on intra- and inter-individual model evaluation and merged them with the section Final model evaluation and interpretation in the Methods section.

Comment 9: Line 309, it is unnecessary for a reader to know both values of reaction time, just use the one in ms. Details such as this one make the paper difficult to read. Please choose the information such that is essential for the paper findings.

Response 9: Although we agree that just one scale (e.g., in ms) would facilitate the reading experience, we think that it is necessary to keep both scales (i.e., log-ms and ms) for the following reasons: a) the statistical tests are based on the log-ms scale, and b) we compare the predictive performance of our EEG model to other models from the literature for which researchers have reported values in ms. Therefore, we would prefer to keep both scales in our manuscript.

Comment 10: For ANOVA, please write dependent and independent variables, and if it is 1-way or not... Was an Anova performed for EEG model as well? Here, there is only the RT model?

Clarify whether the RT model is intra-individual as well, and what does the average RT represent exactly (avg per how many epochs, give example).

Response 10: We exclusively performed an ANOVA on the EEG model as we were mainly interested in neural correlates of mental rotation and used the RT model rather as a baseline model. To further specify the applied ANOVA, we have modified that section (lines 353-359): “To evaluate whether the intra-individual predictive performance of the EEG model varied as a function of angular disparity, we performed a one-way repeated measures ANOVA including participant as a random effect and angle as a fixed effect nested within participant. The ANOVA revealed that there was no significant difference in the dependent variable, MAE, between the levels of the independent variable, rotational angle (F(3, 117) = 0.22, p = .88).”

Comment 11: In the result section, the authors explain inter-individual method that should be explained in the methods section, before the results, and with examples.

Line 334 might be an error, as there should be 39 train participants for the one remaining to be tested? It seems like there is an overfitting here. Again, not very clear without examples. Do the authors perform LOSO (leave one subject out) evaluation?

Response 11: We have moved the explaining section to the Methods section and added examples to clarify our procedure (lines 299-314): “Similar to the intra-individual model evaluation, we extracted for each participant (“train participant”) features using reaction time parameters and spatial filters from their training set. After individually training the EEG model with the cross-validated value for lambda, we evaluated for every other participant (“test participant”) the inter-individual prediction performance (i.e., MAE) based on the samples from the test participant’s hold-out set [23]. In the first iteration, we trained the EEG model with data from participant 1, the train participant, and evaluated its predictive performance 39 times (i.e., using the data from each test participant once: participant 2, 3, …, and 40). In the second iteration, we trained the EEG model with data from participant 2 and evaluated its performance with the data from participant 1, 3, 4, …, and 40. This procedure continued until we had used all participants once for training. Finally, we averaged for each train participant the MAE measured in the hold-out sets from the remaining 39 test participants. To evaluate the inter-individual prediction performance of the EEG model, we compared for each participant the MAE of the participant’s EEG model using (a) the same participant’s hold-out set and (b) the hold-out sets of all other participants (whose MAE we then averaged).”

Comment 12: Here, the authors also mention individualized "pre-processors" which were not mentioned earlier. I suggest to either delete this part or make it more clear (and write it in the right section), with examples. Please also write the motivation behind this comparison, and maybe choose one meaningful result.

Response 12: We have updated the Methods section to now introduce the motivation behind the impact of “pre-processors” (lines 314-317): “To inspect the impact of person-specific reaction time aggregates and spatial filters (“pre-processors”) on the inter-individual predictive performance of EEG models, we evaluated the inter-individual performance additionally using the test participant’s pre-processors for feature extraction.”

Comment 13: In the Discussion there are again explanations of the methods; It is fair to give a few short reminders but there is no need for detailed and repeated explanations that should be stated in the methods sections.

Response 13: We have removed the detailed statements from the beginning of the Discussion section and shortened it to (lines 438-445): “The present study designed and evaluated a person-specific machine learning approach to estimate the contribution of EEG features to predicting the latency of correct responses in a mental rotation task. Using established methods for modelling mental processes based on neural activity, in combination with the widely used mental rotation task capturing visuospatial performance, we successfully created person-specific models which accurately predicted reaction times based on that person’s EEG activity and to a lesser degree based on another person’s EEG activity. Additionally, we explored the contribution of the various EEG features to the final prediction.”

Comment 14: Line 442, the authors write about relying on the average RT again which remains confusing. How are average RTs predicting RTs?

Response 14: We have re-formulated this as follows (lines 287-292): “To evaluate the intra-individual prediction performance of the EEG model, we measured for each participant the MAE for (a) the “EEG model” which predicted reaction times based on the EEG features in the participant’s hold-out set and (b) the “RT model” which estimated for the hold-out set the average, standardized reaction time per angular disparity (i.e., 0). The RT model did not take new data from the hold-out set into account and exclusively generated the new value based on the data from the training set.”

Comment 15: The limitations are nicely written, however the Discussion is again very long and it seems repetitive with the rest of the paper; please shorten the paper overall.

Response 15: We have shortened the limitations and removed repetitive statements.

Comment 16: The paper simply needs re-writing but seems to be correct; I shall know better when it is more understandable. The only true issue is this sliding cross validation method for lambda tuning; I am not sure I understand why this method was chosen, and if it is completely valid...

Response 16: We thank the reviewer for the helpful comments and hope to have addressed the raised issues. For a detailed response to the sliding cross-validation, we would like to refer to our response to comment #4.

Comments by reviewer 3

Comment 1: This manuscript reports a compelling investigation into the potential of an individually tailored machine learning approach to identify EEG patterns of neural activity during mental rotation. The introduction provides extensive background on clinical groups associated with changes in the performance of mental rotation tasks. The final section presents the ultimate goal of this research as “the identification of the neural substrates may represent a promising approach for neurophysiological stimulation studies to finally restore impaired behavioural functionality.”

The proposed machine learning model is based on spectral EEG features (10 frequency bands) and the target is based on the reaction times of the correct answers. The definition of the target vector takes into account the specificity of the task (rotation angle and reaction time). The hyperparameter of the model is optimized in a cross validation framework. The trained model is tested in an hold-out set and intra-individual prediction performance assessed using Mean Absolute Error. Feature importance is analyzed based on the SHapley Additive exPlanations (SHAP) method. The authors evaluated two models: the EEG model (intra-individual), which assessed individual prediction performance, and the RT model, which predicted the average, standardized reaction time per rotation angle (group-level).

The paper's main contribution is a significant difference between intra-individual model and an inter-individual evaluation regarding the prediction of the reaction time of a mental rotation task.

The paper is well-written, but may improve its comprehensibility. The methods are straightforward but rather puzzling in some aspects. The results are well-framed in the literature and pave the way for future studies.

Several aspects caught my attention, and I would like the authors to address and discuss them in a revised version.

Response 1: We thank the reviewer for their comments.

Comment 2: Why not instruct the participant to follow the same strategy (response as soon as possible)? (“We instructed participants to choose a strategy for responding (i.e., either slower and more accurate or faster and less accurate) and to stick to it throughout the task.”) This is critical since one of the variables of interest is the reaction time. Is there a relation between the number of epochs available after preprocessing and the strategy followed by each participant?

Response 2: Due to the imposed time limit, our implementation of the mental rotation task requires participants to balance the speed-accuracy trade-off. During some preceding practice trials participants became familiar with the task and its difficulty. We expected participants to be more comfortable in following the strategy they selected for responding than following a strategy instructed by the experimenter (e.g., respond as soon as possible) which may reduce non-specific alterations in task-ongoing EEG activity (e.g., related to induced stress). Thereby, we prioritized within-person consistency (in response strategy) over between-person consistency. To acknowledge that this may influence the generalizability of one participant’s EEG model to another participant’s data, we have added this limitation to the discussion section (lines 522-525): “While we expected to reduce within-person variation of reaction times unrelated to mental rotation by instructing participants to choose a response strategy, this may have resulted in inter-individual differences of the response strategy, partly impacting the generalizability of person-specific EEG models.”

Unfortunately, we did not record the strategy used.

Comment 3: It is not clear when was the window of 500ms selected - after the trigger corresponding to the presentation of the two images? why not using 500ms before the button press preparation (from -700ms to -200ms)? This would correspond to the final stage of decision making. Comparing the classification results between both windows would also inform regarding the importance of both stages.

Response 3: We have added more details on the specification of the time window (lines 214-216): “We chose the initial 500 over the last 500 ms per epoch as EEG microstates indicate that the initial period contains processes related to the encoding of visual information and mentally rotating an object [39].” 

Additionally, we have added the comparison of different time windows to our discussion (lines 554-557): “Finally, comparing different time windows (e.g., initial 500 ms vs last 500 ms) informed by EEG microstates analysis may reveal how spectral and spatial EEG features differ between stages of cognitive processing (e.g., encoding of visual information, mental rotation, decision making) [39].”

Comment 4: The authors present the hyperparameter tuning after epoch extraction. However, I believe that it would be easier to understand after feature extraction, as these feature sets will be used in model training during the cross-validation procedure.

Response 4: We have followed your advice and moved the hyperparameter tuning section.

Comment 5: The definition of two evaluation targets (EEG model and RT model) should be addressed earlier as well as the objectives of each model. The EEG model is very nicely explained. However, the RT model is quite puzzling to me - I suggest the authors to describe it independently as it is not clear to me which was the training and the target (as far as I understand, it represents a group analysis - the same features and target - instead of an individual one, i.e., the training of the RT model includes all participants?).

Response 5: For intra-individual comparisons, the RT model generates for each participant’s hold-out set the average reaction time of the same participant’s training set. As we standardized reaction times, the RT model exclusively generates zeros. We hope to have clarified the definition in our modified version of the text (lines 287-298): “To evaluate the intra-individual prediction performance of the EEG model, we measured for each participant the MAE for (a) the “EEG model” which predicted reaction times based on the EEG features in the participant’s hold-out set and (b) the “RT model” which estimated for the hold-out set the average, standardized reaction time per angular disparity (i.e., 0). The RT model did not take new data from the hold-out set into account and exclusively generated the new value based on the data from the training set. For the EEG model, we extracted the features from the last 25% of epochs using the same reaction time parameters (i.e., means and standard deviations per angular disparity) for standardization of reaction times and the same spatial filters as estimated in the initial 75% of epochs. Next, we estimated (a) reaction times with the trained model and the new data from the hold-out set (EEG model) and (b) average reaction times per angular disparity (RT model).”

Comment 6: Line 306: Considering the results of the intra-individual and RT model, it is rather strange that the results and significant - add statistical details of the test.

Response 6: We have reported the statistical details in the last sentence of the methods section. As we used bootstrapped paired t-test, we cannot provide degrees of freedom. However, we have added the results of a paired t-test without bootstrapping for a comparison (lines 339-341): “As a comparison, when we re-ran the analysis without bootstrapping, the difference remained significant (t(39) = -4.75, p < .001).”

Comment 7: One alternative to inter-individual model evaluation would be to bootstrap across participants, i.e. train the model with 39 part and test in the remaining one (repeating this process for all participants). What is the comment of the authors regarding this hypothesis?

Response 7: To comment on this hypothesis, we have added the following statement to the discussion section (lines 482-487): “Taking individual topographical patterns into account was of major importance in generalizing one person’s model to the data from another person. However, the EEG model continued to predict reaction times more accurately for the same person it was trained on than for another person suggesting that leave-one-participant-out analyses may fail to uncover topographical patterns beneficial for within-person predictions.”

Comment 8: The model presented in line 344 (second approach to inter-individual evaluation) is quite puzzling. As far as I understand, the features used to train the model and to test it are not the same (different participants may have different parameters).

Response 8: Yes, your understanding is correct. To clarify the approach, we have added the following sentence to the methods section (lines 314-317): “To inspect the impact of person-specific reaction time aggregates and spatial filters (“pre-processors”) on the inter-individual predictive performance of EEG models, we evaluated the inter-individual performance additionally using the test participant’s pre-processors for feature extraction.”

Comment 9: page 5, line 84: Additional information on the features considered by the classifiers. Additionally, please re-phrase “achieved a mean absolute error between 100 and 600ms”

Response 9: We have modified that sentence (lines 89-91): “Furthermore, machine learning approaches modelling behavioural responses as a function of the preceding EEG power in four to ten frequency sub-bands performed with a mean absolute error between 100 and 600 ms [21,22].”

Comment 10: line 190: “an insufficient amount of data was recorded”. This happened due to an early response (<700ms) of the participant? please address this in the manuscript.

Response 10: We revised this sentence as follows (lines 207-210): “From these epochs we removed those during which (a) the fixation cross appeared, (b) the given response was incorrect, and finally (c) an insufficient amount of data was recorded (i.e., when participants responded after less than 700 ms).”

Comment 11: The presentation of the SHAP values for a representative participant (e.g. figure 4) is not necessary in my opinion and could be presented in suppl. materials. Figure 5b is also quite strange as no discussion/interpretation is presented on this results.

Response 11: We think that the SHAP values for a single participant are important to demonstrate that the direction (i.e., positive or negative) of the EEG feature – reaction time relationship differs between features. For Fig 5b we have added the following statement to the discussion (lines 470-473): “Overall, frequency bands close to one another were more similar regarding their importance for the prediction of reaction times than distant bands (Fig 5b) which supports previous work based on highly similar spatiotemporal characteristics between sub-bands [43].”

43. Cohen MX. A data-driven method to identify frequency boundaries in multichannel electrophysiology data. Journal of Neuroscience Methods. 2021;347: 108949. doi:10.1016/j.jneumeth.2020.108949

---

## [Decision Letter · Decision Letter 1]

11 Oct 2023

PONE-D-23-18489R1Estimating person-specific neural correlates of mental rotation: A machine learning approachPLOS ONE

Dear Dr. Uslu,

Thank you for submitting your manuscript to PLOS ONE. After careful consideration, we feel that it has merit but does not fully meet PLOS ONE’s publication criteria as it currently stands. Therefore, we invite you to submit a revised version of the manuscript that addresses the points raised during the review process.

We look forward to receiving your revised manuscript.

Kind regards,

Humaira Nisar

Academic Editor

PLOS ONE

Reviewers' comments:

Reviewer's Responses to Questions

**Comments to the Author**

1. If the authors have adequately addressed your comments raised in a previous round of review and you feel that this manuscript is now acceptable for publication, you may indicate that here to bypass the “Comments to the Author” section, enter your conflict of interest statement in the “Confidential to Editor” section, and submit your "Accept" recommendation.

Reviewer #1: (No Response)

Reviewer #2: All comments have been addressed

2. Is the manuscript technically sound, and do the data support the conclusions?

Reviewer #1: Partly

Reviewer #2: Yes

3. Has the statistical analysis been performed appropriately and rigorously? 

Reviewer #1: N/A

Reviewer #2: Yes

4. Have the authors made all data underlying the findings in their manuscript fully available?

Reviewer #1: No

Reviewer #2: No

5. Is the manuscript presented in an intelligible fashion and written in standard English?

Reviewer #1: No

Reviewer #2: Yes

6. Review Comments to the Author

Reviewer #1: The authors partially answered to my concerns. In particular,

(Comment #2) Please, describe the mental rotation task and its application also in the Introduction

(Comment #3) Please, deepen the usufulness of your approach on the prediction of RT in the mental rotation task

(Comment #8 #10) Please, avoid repetions in the paper and describe in a chronological order your data processing. Please, specify ALL the details that the reader needs to know to reproduce your pipeline.

Reviewer #2: The authors have thoroughly replied to all my concerns and comments. There is one minor modification I could not see, that is the Figures seem to not have changed, maybe I have trouble seeing the difference. Could the authors kindly indicate the changes made in the Figures?

7. PLOS authors have the option to publish the peer review history of their article (what does this mean?). If published, this will include your full peer review and any attached files.

Reviewer #1: No

Reviewer #2: No

---

## [Author Response · Author response to Decision Letter 1]

17 Nov 2023

Reviewer 1

-----------

Comment 1: 

The authors partially answered to my concerns. In particular,

(Comment #2) Please, describe the mental rotation task and its application also in the Introduction

Response 1: 

We have added a description in the introduction (lines 34-35): “It involves the judgement of rotational invariance based on objects rotated in space.”

Furthermore, we think to have addressed the application of the mental rotation task in the introduction:

1 (lines 33-34): “The mental rotation task has frequently been used to invoke complex cognitive processes including visuospatial representations and visual working memory.”

2 (lines 54-73): “In patients showing progressive behavioural and neurological decline, as for example in those with Huntington’s disease, subtle impairments in visuospatial abilities found in early stages transition to significant differences in symptomatic stages when compared to healthy controls [8]. […] In summary, mental rotation in individuals with clinical conditions has been found to be altered; the identification of neural substrates, therefore, may represent a promising approach for neurophysiological stimulation studies to finally restore impaired behavioural functionality.”

Finally, we described our implementation of the mental rotation task in the methods section (lines 141-157): “As part of the study, we assessed the performance of participants in a computerised mental rotation task with a total of 192 trials [31]. […] We instructed participants to choose a strategy for responding (i.e., either slower and more accurate or faster and less accurate) and to stick to it throughout the task.”

Comment 2:

(Comment #3) Please, deepen the usufulness of your approach on the prediction of RT in the mental rotation task

Response 2:

We have added the following statement (lines 114-117): “To assess the commonly assumed generalizability of neural correlates across participants [2-5,11] and to improve the predictive accuracy of reaction times using EEG signals, we present a machine learning approach to extract person-specific neural correlates of mental rotation.”

Comment 3:

(Comment #8 #10) Please, avoid repetions in the paper and describe in a chronological order your data processing. Please, specify ALL the details that the reader needs to know to reproduce your pipeline.

Response 3:

1. Avoid repetitions

Reviewer 2 had raised the issue of repetitions in the first review round and requested additional examples:

>>The paper is written in proper English, however it is long to read as it has many repetitions and a lack of examples. It seems that the authors give many details on results that are not essential to the final findings. For the sake of readability, I would suggest to shorten some technical details and results (and their repetitions) that do not bring clarity to the paper. <<

We assume to have resolved the issue of repetitions in the first review round:

>> We thank the reviewer for these suggestions. We have removed the repetitions (see our response to comments #8, 11, 13, 15) and added some examples (see our response to comments #7, 11). Furthermore, we have shortened the results section and its repetition in the discussion (see our response to comments #8, 11, 13, 15). <<

2. Describe in a chronological order your data processing

Reviewer 3 had raised the issue of chronological order in the first review round:

>> The authors present the hyperparameter tuning after epoch extraction. However, I believe that it would be easier to understand after feature extraction, as these feature sets will be used in model training during the cross-validation procedure. <<

We assume to have resolved the issue of chronological order in the first review round:

>> We have followed your advice and moved the hyperparameter tuning section. <<

We have additionally created a table containing the pseudo code of data processing (line 172-173): “The chronological order of processing steps is summarised below (Table 1).”

3. Please, specify ALL the details that the reader needs to know to reproduce your pipeline.

We have uploaded the source code to reproduce our analyses (lines 171-172): “The associated source code to reproduce the analyses is available at https://github.com/UsluSinan/EEG-correlates-of-mental-rotation.”

Reviewer 2

-----------

Comment 1:

The authors have thoroughly replied to all my concerns and comments. There is one minor modification I could not see, that is the Figures seem to not have changed, maybe I have trouble seeing the difference. Could the authors kindly indicate the changes made in the Figures?

Response 1:

In the revised version, fig 1 (formerly fig 2) includes “EEG data” and “RT” instead of “X” and “Y”. Furthermore, we have removed the equations from the figure.

---

## [Editor Report · Decision Letter 2]

2 Jan 2024

Estimating person-specific neural correlates of mental rotation: A machine learning approach

PONE-D-23-18489R2

Dear Dr. Uslu,

We’re pleased to inform you that your manuscript has been judged scientifically suitable for publication and will be formally accepted for publication once it meets all outstanding technical requirements.

Kind regards,

Humaira Nisar

Academic Editor

PLOS ONE

Additional Editor Comments (optional):

Thank you for revising the manuscript based on author's comments.
---

## [Editor Report · Acceptance letter]

22 Jan 2024

PONE-D-23-18489R2 

PLOS ONE

Dear Dr. Uslu, 

I'm pleased to inform you that your manuscript has been deemed suitable for publication in PLOS ONE. Congratulations! Your manuscript is now being handed over to our production team.

Kind regards, 

on behalf of

Dr. Humaira Nisar 

Academic Editor

PLOS ONE